# A Cloud Microservices Architecture for Data Integrity Verifiability Based on Blockchain

Juan Carlos López-Pimentel [1,*,†] , Luis Alberto Morales-Rosales [2,*,†] and Ignacio Algredo-Badillo [3,†]

1 Facultad de Ingeniería, Universidad Panamericana, Álvaro del Portillo 49, Zapopan 45010, Jalisco, Mexico
2 Faculty of Civil Engineering, CONACYT-Universidad Michoacana de San Nicolás de Hidalgo, Morelia 58000, Michoacán, Mexico
3 Faculty of Civil Engineering, CONACYT-Instituto Nacional de Astrofísica, Óptica y Electrónica, Tonantzintla 72840, Puebla, Mexico; algredobadillo@inaoep.mx
* Correspondence: clopezp@up.edu.mx (J.C.L.-P.); lamorales@conacyt.mx (L.A.M.-R.); Tel.: +52-3313682200 (J.C.L.-P.)
† These authors contributed equally to this work.

**Abstract:** The current digital age, mainly characterized by an economy based upon information technology, demands a data integrity service, even more so because organizations and companies are migrating their services over the cloud. This is not a simple task; it is cumbersome since traditional schemes in databases could be subject to modifications. However, it can be solved using blockchain technology. This paper provides a data integrity verifiability architecture for cloud systems based on blockchain. The architecture provides a mechanism to store events (as logs) within a blockchain platform from any cloud system. Users can then consult data integrity through a microservice, acting as an intermediate server that carries out a set of verification steps within the blockchain, which confirms the integrity of a previously stored log. Our architecture takes advantage of the blockchain strength concerning integrity, providing a traceability track of the stored logs. A prototype system and a case study were implemented based on the proposed architecture. Our experimental results show that the proposed decentralized architecture can be adapted to cloud existing systems that were born without blockchain technology and require a modular and scalable audit characteristic.

**Keywords:** blockchain; integrity; audit; microservices; cloud systems

## 1. Introduction

In recent years, cloud computing has been one of the most important topics in the field of information technology (IT) [1]. Every day, many organizations and companies are migrating their services over the cloud [2]. However, a primary obstacle in moving their systems to the cloud concerns the security and the continuously increasing number of digital crimes occurring in cloud environments [1]. Ensuring data integrity in any system is currently an imperative requirement. Stored data can be vulnerable to manipulation, intentionally or not, even more when placed in the cloud. Data integrity is a fundamental aspect of storage security and reliability [3]. It consists of protecting information against modifications or inappropriate destruction.

Policies, regulations, and secure mechanisms should be developed to protect people from being deceived; cloud forensics is a step forward in dealing with it [1]. A fundamental mechanism for digital forensics is auditing the logs; they are used to keep track of important activities. Logs have been used to provide information about past events and, therefore, show the path of the state of a system [4].

"Ensuring" logs from improper users has been reviewed by many researchers in different areas, such as distributed systems [4,5], cloud computing [6,7], and services [8–11]. These system solutions have been developed to be adapted for TCP/IP internet infrastructures. The internet was built to guarantee communication, not to guarantee

information security properties. Thus, TCP/IP-based solutions layered on the internet have raised significant trust and privacy concerns.

Blockchain seems to be a suited technology for dealing with cloud system data integrity, because blockchain is characterized by features, such as transparency, traceability, and security [12]. These features make the adoption of blockchain attractive toward enhancing information security, privacy, and trustworthiness in very different contexts. Firstly, this technology was developed to improve integrity in the Bitcoin network [13]; then, it gave way to other cryptocurrencies. Recently, blockchain and the smart contract extension have permitted considering other use cases, such as the supply chain, Internet of Things, healthcare systems, digital right management, insurance, financial systems, and real estate systems [14]. These implementations share a very important characteristic related to trust—a crucial element of blockchain technology [15,16].

Blockchain applies cryptography and hash functions to chain the blocks, along with a decentralized structure, making it very difficult for third parties to manipulate the information compared to traditional databases. Its potential is enormous in information transactions, such as cloud storage. Blockchain technology itself already has the strengths of including logging and traceability mechanisms. However, the reality is that most organizations were born without it, and their storage information is executed via traditional databases (off-chain). A complete migration could imply very high economic costs since systems with high information content would require too much computational costs deploying their transactions on the blockchain network [17].

In this article, blockchain technology is used as a repository to store all events that can arise in an external system, ensuring its integrity. We provide an audit mechanism, propose a hybrid solution architecture on a permissionless blockchain with smart contract support. Each event, before being stored, is transformed in a log containing the attribute tags that respond to the following questions: how was the event generated? Who generated it? What included the event? Where was it generated? When was the event generated? The log is stored in the blockchain by executing smart contracts. This will allow validating and knowing the traceability of any data modification, guaranteeing if the information provides integrity or not, i.e., knowing the possible alterations by third parties.

On the other hand, following the new technological demands, in terms of software development, we identified that the microservice approach offers an evolutionary step concerning monolithic ones. Thus, we provided an architecture following the microservice approach. However, a microservice system design remains challenging since it can be independently deployed, has lower coupling, and is self-contained. For cloud applications development, microservice allows more freedom to evolve and requires a mechanism to inform others about what has changed or to track past events in the system. With the highly decentralized development and design of microservices, it becomes difficult to maintain a centralized architectural design reference [18]. As a result, the system might be vulnerable to data integrity attacks. Hence, this paper aims to provide a decentralized architecture with two general services. Firstly, a data integrity mechanism in which all events (records) triggered by a cloud system are stored within a blockchain in a hashed form. Secondly, a verifiability process that checks the data integrity. The mechanism was built through a microservice architecture to be integrated with any system as a service.

We implemented our architecture, and our experiments show that it can be successfully suited within other cloud systems. Our main contributions are listed as follows:

- A mechanism for auditing issues storing logs from the initialization phase of a system until the execution of user transactions guaranteed by blockchain strengths concerning integrity.
- A consulting mechanism that provides certainty about the integrity of the logs stored in the blockchain.
- Traceability of each log transaction carried out by the corresponding users of a cloud system in distributed environments.

- An auditing mechanism where cloud systems (following the microservice architecture) can plug it.

Our proposal falls in a design science research in the information systems area [19]; specifically, in the software engineering topic. The methodology of our work follows that proposed in [20], commonly applied in engineering. Such a methodology comprehends the following steps: (a) Motivation and problem identification; (b) Objectives; (c) Design; (d) Development; and (e) Demonstration and validation.

Following the methodology described previously, we structured the paper as follows: (Step a) is deepened in Section 2, which also grounds some technologies used throughout the document and some related works; Section 3 raises the objectives (Step b) and describes our general architecture (Step c) and main components; Sections 4 and 5 detail the design of our architecture (continuing Step c); Section 6 explains our prototype (Step d), together with a case study shown in Section 7; some demonstrations and validations (Step e) are provided in Section 8. Finally, we outline our discussion and conclusions in Section 9.

## 2. Preliminaries and Related Work

The amount of data currently generated worldwide does not have a direct comparison with the data generated in the past since users create and store more data than ten years ago. For example, a single user currently solely provides large amounts of data per day in social networks. The large amounts of data are often stored in the cloud. Thus, one of the current high-demand issues is data integrity, especially for auditing matters.

On the other hand, companies are migrating their processes to the cloud, causing the implementation of new strategies in the software development process. This section starts contextualizing the evolutionary requirements of the automation systems highlighting what worked in the past does not necessarily work today. Then, we approach some works that focused on the subject of data integrity in cloud systems and those focused on blockchain.

### 2.1. From Programming Office Systems to Microservice Architecture

Office systems, reference [21], were the first good solutions for the automation of organizations. Over time, these systems became fragmented subsystems; then, an excellent evolution was client–server systems. Organizations could obtain information in real-time from centralized databases, which represented significant advances in software development. A client–server system (denoted in the literature as monolithic architecture) is a way to build a complete software as a unit and the central server has several responsibilities and does mostly everything. Monolithic architecture is still valid for small projects, but when organizations or team groups become more extensive, client–server systems have been shown to not be enough, reference [22].

Microservice architecture is a good solution to solve the monolithic problem and it recently has become more popular. It is a software architecture design that offers the following characteristics: modularity, scalability, integration of heterogeneous and legacy systems, and distributed development, reference [23]. These characteristics are highly required when designing a system that will be mounted in the cloud. Although the microservice system design remains challenging, it has become the leading design for cloud native systems; since it provides benefits, such as faster deployment cycles, better scalability, and good separation of concerns among services, reference [18].

### 2.2. Data Integrity in Cloud Systems

Cloud computing can be perceived as a collection of computing resources providing shareable infrastructure through the internet,which are accessible and available everywhere, reference [24]. The name was coined by Mr. Ramnath Chellappa in 1997, reference [25], which agree with [24,26].

Cloud computing is defined by the National Institute of Standards and Technology (NIST) as a model for enabling convenient, on-demand network access to a shared pool of configurable computing resources (e.g., networks, servers, storage, applications, and

services) that can be rapidly provisioned and released with minimal management effort or service provider interaction, reference [27]. Applications of cloud computing are mailing services to storage, document processing, hosting services, image processing, and video streaming [26].

The reality is that cloud computing right now is one of the most trending topics in the field of information technology (IT) in recent years [1]. Companies, such as Amazon, Google, and Microsoft, have enhanced their services to provide cloud environments for their customers. Thus, every day, many organizations and companies are migrating their services over the cloud, since the benefits of cloud computing encompass improved efficiency, flexibility, and reduced infrastructure costs, reference [2].

However, there exist some concerns when uploading systems to the cloud, such as security, privacy, power efficiency, compliance, and integrity [1,28]. Preserving confidentiality and integrity in distributed systems and cloud services is also a key problem. The importance of confidentiality and data integrity in cloud systems is imminent. Confidentiality refers to restricting access to information, at least, it is authorized to access it. Data integrity aims to protect information against intruder modifications or software bugs.

Traditional techniques used for data integrity and verification for cloud storage are already provided in the literature [29]. Man et al. [30] argue that traditional systems (only off-chain) remain inefficient, antiquated, leading to critical data omissions, security vulnerabilities, and even corruption. We explore a newer technique, the combination of data integrity with a blockchain platform.

### 2.3. Data Integrity with Blockchain

Blockchain, firstly introduced by Satoshi Nakamoto [13] in the design of Bitcoin, is described as a distributed database, where the information contained is allocated in blocks; each block is chained with its previous one by adding the hash address of the previous block, among other data, establishing a chain. Each block can also include smart contracts, a set of computer instructions that are triggered to run specific tasks. Blockchain is an excellent option for data integrity; it is resistant to malicious data modification.

Some initiatives have explored the use of blockchain to provide integrity in audit issues. Suzuki, et al. [31], proposed a scheme using blockchain technology as a request-response channel for a client–server system in a control access application to record both client requests and the server responds in an auditable manner. Ahmad et al. [32] argued that traditional log systems are vulnerable and subject to a series of attacks; they searched for a system able to avoid audit logs being "tamped" by adversaries; then, they proposed a system called BlockAudit, which used Hyperledger blockchain. There are other works based on Hyperledger, focusing on auditing data integrity using schemes involving a third-party auditor [33]. PengCheng et al. [34], use a mobile agent technology to deploy distributed virtual machine agent model in the cloud; the machine agent enables to cooperate to store data, then they built an integrity protection mechanism with their virtual machine proxy model; they trust in the security mechanism of the blockchain technology to improve the performance of the cloud computing concerning secure storing and secure computing. Even though there are works focused on blockchain in data integrity aspects, we aimed to provide a data verifiability architecture that can be integrated with other systems, in particular with cloud-based.

A closely related work is presented in [35], an audit mechanism that saves all the events of a supply chain in a blockchain in a hashed form; then users of the supply chain system can query integrity, data provenance, and traceability from a blockchain through an intermediate server that establishes communication between blockchain and the supply chain. Recently, *RootLogChain* was proposed in [36], it is an audit mechanism that is built upon a security protocol to create a root user in a blockchain, and from there, all root events are stored as logs within a blockchain.

## 3. The Architecture Model

This work provides an architecture able to store all events (logs) within a blockchain in a hashed form; then, we give a verifiability process to check the data integrity. This architecture can be adapted to cloud systems that generate HTTP events. These events must be triggered to the architecture to be saved.

Figure 1 illustrates an outline of our model. The Cloud System, denoted as CS, is illustrated in the left part of Figure 1. CS must establish communication with The Verifiability Blockchain Service Interface, abbreviated as VBSI, for both to store each event that occurs internally within CS and to provide reliability by means of a consulting process. VBSI is an intermediary between CS and the blockchain part. Blockchain acts as a software connector [37], where all events are registered. The following subsections outline the general operability of these three parts: CS, VBSI and the blockchain.

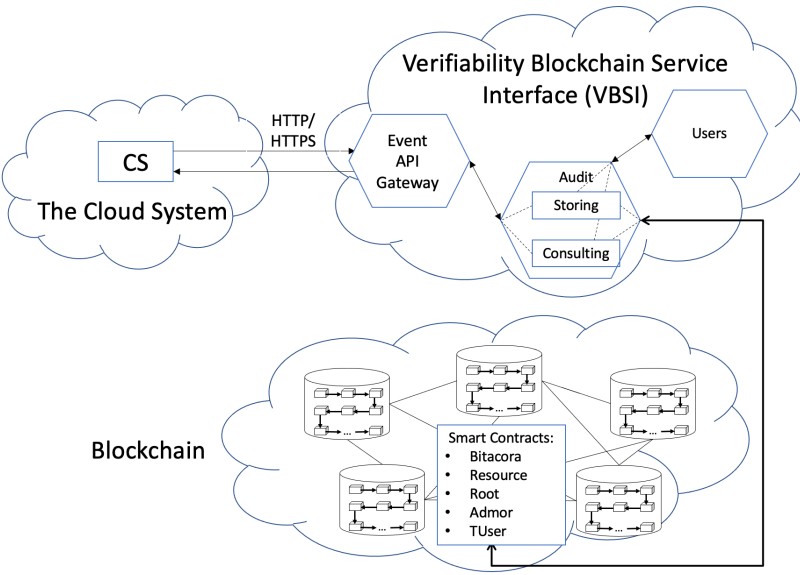

**Figure 1.** General Model: interaction among The Cloud System, The Verifiability Blockchain Service Interface and the blockchain.

### 3.1. The Cloud System

From the final user perspective, cloud computing can be divided into two parts: the client and the server [38]. The client part comprises the software used by the final user. The user does not have to install complex software to use services provided by the cloud; usually, a web browser, mobile application, or terminal are used. The server side has to do with the back-end, it is an abstract layer with the hardware and the connectivity; it means various computers, servers, and data storage.

On the other hand, according to Zhang et al. [39], the cloud computing architecture consists of four layers: the hardware, the infrastructure, the platform, and the application. The hardware layer, typically implemented in data centers, manages the physical resources such as servers, routers, switches, power, cooling systems, etc. The infrastructure layer shares physical resources using virtual technologies, such as VMware, XEN, and KVM. The platform layer consists of operating systems and application frameworks. The application is the highest layer consisting of the cloud applications; it is the view of the final user.

In the case of our model, we locate The Cloud System in the platform layer according to cloud computing architecture of Zhang et al. [39]. The Cloud System will establish communication with VBSI using the TCP/IP protocol. In particular, it will be over the hypertext transfer protocol (HTTP) [40], and its secure version HTTPS [41]. The methods that The Cloud System will be requesting to VBSI will be GET, POST, PUT and DELETE.

### 3.2. Microservices of VBSI

Considering that our model might be implemented as an extension in systems requiring audit data characteristics, we designed it following a software scheme as a service using a microservice architecture. This architecture was introduced as a new alternative, reference [42], over the monolithic approach. Microservices were developed as small, well-defined purposes, and autonomous services, deployed independently [23,43]. Currently, organizations tend to upload their services to the cloud, and this architecture is usually recommended.

Figure 1 illustrates three general microservices: *Event API-Gateway* , *Audit* and *Users*, as illustrated at the top-right of Figure 1. VBSI is the interface of our model, dedicated to interacting with The Cloud System. CS sends HTTP (respectively HTTPS) request methods to trigger events. These events are received by the *Event API-Gateway*, which assorts the events and resends to one of the following types of services: (a) Storing data (it involves submitting, updating, or deleting information) in a hashed way; or (b) Consulting information for audit issues. Any of these services must check the suited permissions with the *Users* service. The answer sent to The Cloud System depends on the processed operation in the blockchain. Details of the global mechanism of VBSI will be explained in Section 4.

### 3.3. Participants as Smart Contracts

The *blockchain* part stores events in order to validate the information stored in the The Cloud System. Our proposal of including blockchain as a verification procedure coincides with [35,44]. Each event in the system is marshaled in hashing data to be stored by smart contract named Bitacora.

To execute, this smart contract is required to be called by a specific type of user. We created three types of users (root, administrator and transaction users), these type of users are represented in the blockchain as smart contracts. The general overview is as follows:

- Root: It is a crucial role, which is formed at the time that the system is initiated. Our model trusts in the root, and a smart contract Root is created the first time when the architecture is started. Hence, all subsequent operations are chained to this contract. The creation of the root is carried out following [36].
- Administrator: It is created by the root. An independent smart contract is created for each administrator. In our model, the administrator can add TUsers.
- TUser: When an administrator adds a new user, creates a TUser smart contract, and also generates a Resource smart contract. TUser is a transactional user who can store hash data in the blockchain through a smart contract called Resource.

Details of the global mechanism of the smart contracts will be explained in Section 5.

## 4. The Verifiability Blockchain Service Interface (VBSI)

VBSI interacts with two external entities: The Cloud System and the blockchain. The interaction with The Cloud System is performed through the Event API-Gateway, while the interaction with the blockchain is carried out by the *Audit* microservice. Internally, all operations must be "permissioned" by a microservice, so-called *Users*. In the following sub-sections, we will explain them in detail.

Table 1 gives a general abstract of some notations and abbreviations that can be found throughout the paper. The last column pinpoints the reference formula where it can be found.

**Table 1.** Descriptions and references of some abbreviations and formulas.

| Abbreviation | Description | Section Explained | Formula |
|---|---|---|---|
| CS | The Cloud System | Section 3.1 | |
| VBSI | The Verifiability Blockchain Service Interface | Sections 3.2 and 4 | |
| Audit | Verifiability Blockchain Micro-Service | Sections 4.2 and 4.3 | |
| $Ev$ | Event's type | Sections 4.1 and 4.2 | (1) |
| m | Message sent to VBSI to be stored in the blockchain | Section 4.2 | (2) |
| $A$ | Sender | Section 4.2 | |
| $B$ | Receiver or Target | Section 4.2 | |
| $D$ | Data to be stored in the blockchain | Sections 4.2 and 4.4 | (2) |
| $Hx$ | Hashed message | Section 4.2 | (2) |
| Hash(m) | Hash function on message m | Section 4.2 | (3) |
| $To$ | Token | Section 4.4 | (4) |
| $R$ | Audited answer compound by $\{\!\![To, A_{tr}, A_{sc}]\!\!\}$ | Section 4.2 | (5) |
| $Rp$ | Details generated in the blockchain | Section 4.3 | (6) |
| $Res$ | Receipt answer after a consulting process in the blockchain | Section 4.3 | (7) |
| $P_d$ | Personal data | Section 4.4.1 | (8) |
| $b$ | Boolean answer denotes if a user can or not execute an event | Section 4.3 | (9) |
| $D_p$ | Set of permissions | Section 4.4.1 | (10) |
| $U_t$ | Type of user | Section 4.4.1 | |
| $Gas$ | Gas required to execute a transaction | Section 4.4.1 | |
| $R_S$ | Resource that a user has access | Section 4.4.3 | |
| $K_p^+$ | Public key address | Section 4.4.1 | |
| $A_{tr}$ | Transaction address | Sections 4.2, 4.3 and 5.2 | |
| $A_{sc}$ | Smart contract address | Sections 4.2, 4.3 and 5.2 | |

*4.1. Event API-Gateway*

This follows the API-Gateway architecture of Gadge et al. [45]. The Event API-Gateway is the main controller of the backend. The Event API-Gateway is a server that is the single entry point into the system. It receives events requested by The Cloud System and emits a comeback answer. Such events are denoted as $Ev$ and involve the following CRUD (Create, Read, Update and Delete) operations. These operations are HTTP methods (POST, GET, PUT and DELETE, respectively). It means that an $Ev(\text{m})$ is a polymorphic function, hence, it could refer to any of the following forms:

$$Ev(\text{m}) = \begin{cases} POST(\text{m}) \\ GET(\text{m}) \\ PUT(\text{m}) \\ DELETE(\text{m}) \end{cases} \tag{1}$$

Table 2 represents how the Event API-Gateway receives an HTTP (HTTPS, respectively) request method, and it specifies what is the corresponding interface played in the Audit service. For example, if a Create operation is requested, it is carried out using a POST method, then Storing interface is executed, meaning that a new operation is requested. In

case an Update or Delete operation is requested, Audit service interprets it as an Update operation since no element is deleted.

**Table 2.** CRUD operations sent via HTTP methods and the interpretation in the blockchain by services consulting and storing.

| | Event API-Gateway | | | | | Audit | | |
|---|---|---|---|---|---|---|---|---|
| | Type of Transaction | | HTTP(S) Methods | | | Consulting | Storing | |
| t | CRUD Operation | GET | POST | PUT | DELETE | Read | New | Update |
| 01 | Create | | X | | | | X | |
| 02 | Read | X | | | | X | | |
| 03 | Update | | | X | | | | X |
| 04 | Delete | | | | X | | | X |

The Event API-Gateway communicates internally with *Audit* and *Users* and can also communicate externally with other microservices through another API-Gateway or cloud services directly.

The Event API-Gateway encapsulates the internal system architecture and provides an API that is tailored to each client. One of its responsibilities is to identify the following services: user administration, user authentication, user authorization, audit-storing, and audit-consulting. Other responsibilities are monitoring, load balancing, and caching.

### 4.2. Audit: Storing

Figure 2 illustrates the process of receiving an event and data storing in the blockchain. Let us assume that the Event API-Gateway receives an event $Ev(m)$ with message m to be stored within VBSI. The message is composed of:

$$m = \{|A, B, D, Hx|\} \tag{2}$$

*A* denotes who is sending the message; *B* who is the receiver; *D* a set of data required to be stored in the blockchain; and *Hx* is a hashed of the previous messages, as follows:

$$Hx = \mathsf{Hash}(A, B, D) \tag{3}$$

Token *To* (Section 4.4 for more details) is obtained from *D*:

$$To \in D \tag{4}$$

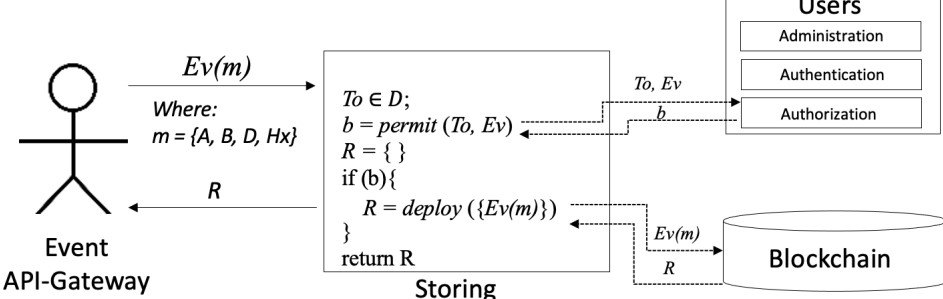

**Figure 2.** Storing operation, illustrating communication between Event API-Gateway, *Storing*, *Users-Authorization* service and the blockchain.

Storing service involves the following off-chain operations:

- Verify with the *Users-Authorization* service if *To* has enough permissions to execute event *Ev*, using Boolean function $b = permit(To, Ev)$. Returning true if it has privileges to store the event in the blockchain; otherwise, it returns false.
- Send the event to the blockchain, using function $R = deploy(\{Ev(m)\})$. Returning *R*, which is a tuple (calculated in the blockchain) of three elements, $\{|To, A_{tr}, A_{sc}|\}$: (a) the token; (b) a transaction address and (c) a smart contract address.

Storing service can work under two scenarios: (a) Creating users; or (b) Executing operations over resources. Both will be explained in detail in Section 4.4.

### 4.3. Audit: Consulting

In order to consult information, Audit has another service, *consulting*, see Figure 3. Let us assume that the Event API-Gateway receives an event $Ev(To, Hx, R)$ to consult a data that previously was stored within VBSI. There, *To* is provided by the authorization service; and

$$R = \{|To', A_{tr}, A_{sc}|\} \tag{5}$$

In this case, *R* is the receipt received when the storing action was carried out (it is the same as explained in Section 4.2); note that $To'$ was the token id approved in the transaction $A_{tr}$ for the smart contract address $A_{sc}$.

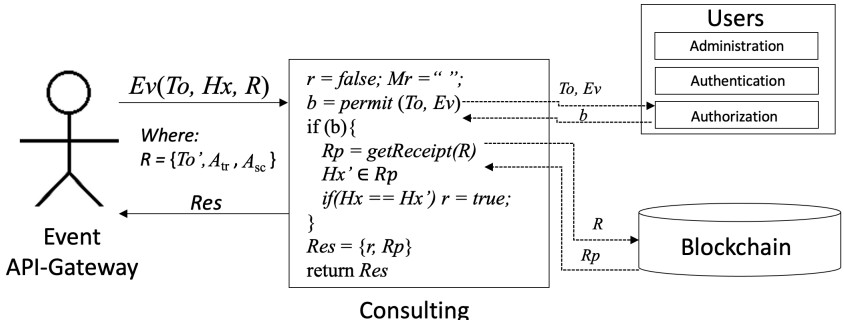

**Figure 3.** Consulting operation, illustrating communication among Event API-Gateway, *Consulting*, and *Users-Authorization* services and the blockchain.

*Consulting* service involves the following off-chain operations:

- Verify with the authorization service that *To* has enough permissions to execute *Ev*; it is used with $permit(To, Ev, R_S)$ function, which is detailed in Equation (9).
- If *b* is true, then it calls to the following remote function:

$$Rp = getReceipt(R) \tag{6}$$

There, *Rp* states for the receipt details consulted with the blockchain and $Hx' \in Rp$ is the hashed information previously stored within the blockchain.
- Then, it is compared *Hx* with $Hx'$, if they are the same, *r* variable is changed to *true*.
- A tuple *Res* is returned, where:

$$Res = \{|r, Rp|\} \tag{7}$$

*Res* can be one of two options according to the previous steps: (a) $r = true$ and the detailed receipt *Rp* when integrity is noted; otherwise, (b) it returns $r = false$ and an empty receipt.

### 4.4. Users

As one can see in Figure 1, the *Users* microservice has direct communication with the *Audit* microservice and the event API gateway. It means that *Audit* or the Event API-Gateway must verify with *Users* whether or not the requests received can be executed or

not. This service acts as an access control service management. Although an alternative solution could be to consume an external system (e.g., software as a service, SaaS) provided by trusted third parties, as some proposed in the literature [46,47]; we preferred developing our solution to take better control of the logs.

Following some proposals in the literature about how to deal with access control management systems [48–50], *Users* microservices provide the following services: administration, authentication, and authorization. Table 3 illustrates all operations. The input column specifies the parameters received, whereas the output column, the answer. Column *storing* pinpoints if the service deserves ($\sqrt{}$) *audit:storing* operation in the blockchain; however, according to the user's convenience, it could be modified. For example, each time a request involves an authentication process, we configured that functions *authenticate*() and *tokenRenovation*() must register a storing operation ($\sqrt{}$) in the blockchain. However, any other implementation could lighten it up and omit it, arguing that it is unnecessary.

**Table 3.** User's microservice.

| Service Name | Function Name | Input | Output | Storing |
|---|---|---|---|---|
| | *userCreation*() | $D, Hx$ | $R$ | $\sqrt{}$ |
| Administration | *userRead*() | $To$ | $J$ | |
| | *userUpdate*() | $D, Hx, R$ | $R'$ | $\sqrt{}$ |
| | *userDelete*() | $D, Hx, R$ | $R'$ | $\sqrt{}$ |
| | *authenticate*() | $P_d, U_t, Ev$ | $To$ | $\sqrt{}$ |
| Authentication | *tokenRenovation*() | $To$ | $To$ | $\sqrt{}$ |
| | *tokenIsValid*() | $To$ | *bool* | |
| Authorization | *permit*() | $To, Ev, R_S$ | *bool* | |
| | *permissions*() | $To$ | *JSON* | |

### 4.4.1. Administration

It provides the four CRUD operations: creation, read, update, and delete. When a request is received to create a user, function *userCreation*$(D, Hx)$ is processed. Where:

$$P_d \in D; To \in D; U_t \in D; K_p^+ \in D; Ev \in D; Gas \in D; D_p \in D \qquad (8)$$

Let $P_d$ be credentials related with personal information of the user to be created, such as *username*, *email*, and a hashed *password*; $U_t$ be the type of user to be created; $K_p^+$ be the public key of the user; *Gas* is the value required to execute the transaction in the blockchain; $D_p$ be a set of permissions that the user will be granted expressed in *JSON* format (Section 4.4.3 explains more about the permissions); and $Hx$ be the hashed of the previous messages. When a user is created, it also requires to create its smart contract, calling a *storing* operation (see Section 4.2), receiving $R$ and it is returned also.

When update and delete are requested (*userUpdate*$(D, Hx, R)$ and *userDelete*$(D, Hx, R)$ respectively), it is assumed that each user has its own smart contract, as it was previously created. These operations require generating new transactions over the existing smart contract, returning a new answer of the blockchain $R'$. These operations internally call for the *storing* service.

Our model follows the idea of [51], which is designed with three general types of users: *Root*, *Administrators* and *TUsers*. The following conditions are followed to build a tree structure:

- The system must have only one root, it follows the idea of [36].
- A public key must be used for only one type of user (validated via off-chain).

- The root user can create administrators (validated via off-chain and blockchain).
- Administrators can create other administrators and TUsers (validated via off-chain and blockchain).
- When a TUser is created, also a *Resource* smart contract is created (instruction sent via off-chain to the blockchain).

### 4.4.2. Authentication

This service provides three operations:

- $authenticate(P_d, U_t, Ev)$; this function receives the user's credentials, type of user and type of event. Returning a token *To* as a result;
- $tokenRenovation(To)$; this function receives a token and returns an updated token. This function is used when a token is about to expire.
- $tokenIsValid(To)$; this function receives a token and evaluates if the token is valid or not, returning a Boolean result.

### 4.4.3. Authorization

This service provides two main operations (9) and (10):

$$b = permit(To, Ev, R_S) \tag{9}$$

This function receives a token, an event, and a resource. The function verifies if the token *To* supplied previously to some type of user has permissions to execute event *Ev* on resource $R_S$. The following is an $R_S$ example in JSON format:

```
{
"resource":        "RootCreation",
"event":          "POST",
"permitAccessTo": {
"Root": "true",
"Administrator": "false",
"TUser": "false"
},
"description":     "Root is the user permitted to create a root"
}
```

This resource is called *RootCreation*, whose event *POST* is permitted only by the root user; the other users are not permitted; a description is also included.

On the other hand, the following function:

$$D_p = permissions(To) \tag{10}$$

receives a token and returns a *JSON* format $D_p$ pinpointing a set of permissions that a user, linked with the token, has available to execute. The following is a permission example $D_p$:

```
{
"stage": "users",
"serviceName": "hasAccess",
"typeOfOperation": "delete",
"nameOfOperation": "deleteMe",
"permitAccessTo": {
"Root": "true",
"Administrator": "true",
"TUser": "true",
"NameType": "Consumer"
},
"description": "Delete personal data"
}
```

which, it is a transaction user with type named "Consumer" can delete its own personal information; the types of users root and administrator have the same privileges.

## 5. The Smart Contracts

Trust in data integrity is one of the primary concerns in computer systems. To remove that worry, we will register all events in a trustworthy architecture, such as blockchain. We consider that events are triggered by different types of users. These users will store events through smart contracts.

We established some particular user roles to limit the operations that each user can execute within the blockchain: Root, Administrator, and a Transaction User (TUser for shorter). TUsers are a generalization of more specialized transactional users. We designed these roles, such as smart contracts.

Our model establishes that any event generated by one of the role users must be registered in a Bitacora and the transactional users have Resources and any action or event over the resources must be registered in their own Bitacora.

### 5.1. Notation

The notation to be used to represent smart contracts will be similar to classes in object-oriented languages. A smart contract is composed of attributes and methods, which can be public (+) or private (−); prefix (*) denotes an attribute internally calculated. To provide encapsulation, all private attributes can be accessed using the corresponding getter method *get[Attribute]()*; for example, let *X* be an attribute, its corresponding getter method would be *getX()*.

A contract also has a constructor, which is a method that uses the same name as the contract and is used to create the contract. Figure 4 illustrates the smart contracts as class diagrams. Private methods can only be called within the contract, as long as public methods are still accessible from other contracts. Abstract methods are not implemented in such a contract. If a smart contract includes at least one abstract method, such a contract is also considered abstract and can only be deployed when abstract methods are implemented. The figure also illustrates some relations of a contract with others, for example: inheritance (*extends* arrow); and use dependency (dotted arrow).

### 5.2. Objectcontract, Log, Bitacora and Root

The base model of the contracts *Objectcontract*, *Log*, *Bitacora* and *Root* were taken from [36], here were used to engage with smart contracts *Admor*, *TUser* and *Resource* as illustrated in Figure 4.

The explanation is as follows:

*ObjectContract* denotes the parent of all contracts. It contains two attributes: contract address $A_{sc}$ and transaction address $A_{tr}$. When a contract is created, these attributes are generated and can be accessed publicly by methods $getContractAddress()$ and $getTransactionAddress()$, respectively. Method $getReceipt()$ is used to obtain the receipt of a transaction. Contract address is used to identify the contract in the blockchain and the transaction address to identify the event.

*Log* inherits from *ObjectContract*, you can see a description of the attributes in Table 4. All of such attributes are stored in the blockchain when method $Emit(Ev)$ is called. Attributes marked with $*$ are auto-generated within the smart contract; the rest are obtained from $Ev$. $Emit(Ev)$ is an abstract method, which must be implemented in a specialized smart contract, such as *Bitacora*. A particular log event can be obtained using $getEvent(Atr)$.

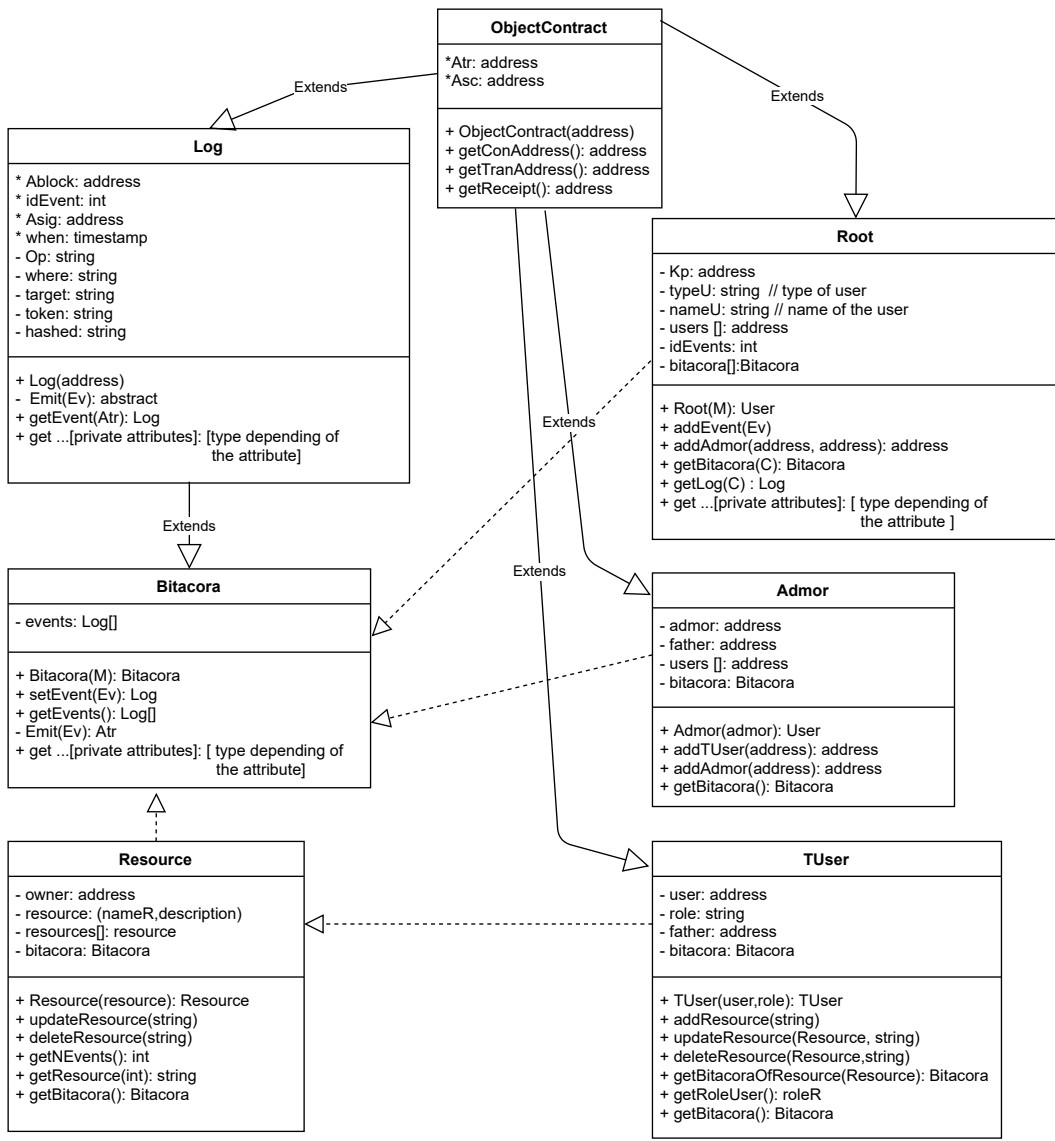

**Figure 4.** The smart contracts illustrated as class diagrams.

**Table 4.** Attributes (*Atr*) and description of Smart Contract Log.

| Abbreviation | Description |
|---|---|
| ∗*Ablock* | Abbreviates Address Block of the blockchain. |
| ∗*idEvent* | Identifier of the current event. This attribute is formed by auto-incrementing. |
| ∗*Asig* | This is a signature hash of the log. |
| ∗*when* | A timestamp *T*, denoting when the event has happened and submitted to the blockchain. |
| *Op* | HTTP methods (GET, POST, PUT or DELETE). |
| *where* | The source of the event |
| *target* | The target of the event |
| *To* | Token identifier to know who has carried out the event. |
| *Hd* | Hashed message that describes more details about the event. |

*Bitacora* inherits all attributes and methods of smart contract *Log*. Events are registered in this smart contract. The constructor *Bitacora*(M) is created when it is called the first time, from M, data are extracted to form *Ev*, and it calls *Emit*(*Ev*), which is a private method, inherited from *Log*. This method stores *Ev* in the *Log*. *setEvent*(*Ev*) is another method, which can add events with information *Ev* by calling *Emit*(*Ev*) to be stored in the *Log*. All log events can be obtained using method *getEvents*().

*Root* smart contract is created executing constructor *Root*(A, AS, m, *To*). The constructor creates its *Bitacora* and the first *Log*. Let A be a user wanting to become a root; AS be a receiver user or target; *To* be a token identifier to identify the transaction; and m is composed of two set of messages $m_1$ and $m_2$. $m_1$ is formed by $\{K_p^+, Gas, credentials\}$, where $K_p^+$ is the user public key, which will be the owner of the contract, and the key address will be used to execute the next transactions; *Gas* represents the cost necessary to perform transactions; *credentials* is related with root's secret information, such as *username*, *email*, and a hashed *password*. With $m_2$ is possible to store extra information of the root.

As a result of executing constructor *Root*, it is generated a blockchain transaction answer *R*, see Formula (5) for more details about *R*.

Smart contract *Root* can add events, using method *addEvent(Ev)*. This method calls *setEvent*(*Ev*) of smart contract *Bitacora*. This transaction generates another transaction, *R*; however, the smart contract address $A_{sc}$ is still the same since this method does not create a new smart contract.

In order to consult a Bitacora and a specific log, smart contract *Root* contains methods *getBitacora*(*R*) and *getLog*(*R*) respectively. Being *R* the answer to the root after triggering an event.

Some modifications were carried out within the smart contract of the Root. We added method *addAdmor*(*address*, *address*), which the Root can add administrators and to store in its private attribute called *users*[]. *addAdmor*(*address*, *address*) requires to specify the smart contract key of the creator (in this case the smart contract of the root in the first parameter) and the key address of the new administrator.

*5.3. Admor, TUser and Resource*

*Admor* is an abbreviation of *Administrator*. An administrator is a special type of user that can add *TUser's* (*addTUser*(*address*)); add other administrators (*addAdmor*(*address*, *address*), specifying the smart contract key of the creator and the key address of the new administrator); and obtain its Bitacora (*getBitacora*()).

*TUser* is an abbreviation of transactional user. In our model *TUser* is the operative type of user of a system able to execute CRUD transactions over some specified module or resource. Neither root nor administrators can deal with resources directly. Thus, tasks related to resources have to do with TUsers.

*TUser* is a generalization type of user, when created (in its constructor *TUser*(*user*, *role*)) is required to pass: the specific role that it will be playing, which can be recovered via *getRoleUser*(); and an address key of the user. *TUser* can create, update, and delete resources by executing methods *addResource*(*string*), *updateResource*(*Resource*, *string*), *deleteResource*(*Resource*, *string*) respectively. TUsers can also obtain their own Bitacora *getBitacora*() and the Bitacora of its Resource *getBitacoraOfResource*().

On the other hand, smart contract *Resource* is only created by a *TUser*, being its owner, and can only be accessed by it. When a resource is created (constructor *Resource*(*resource*)), an *owner* is assigned, in this case, the smart contract of the *TUser* accomplishes this role. The owner of the resource must be the only one that can update it or/and delete it, using its respective methods, see Figure 4.

## 6. Prototype System

The following prototype aims to give a better idea about the architecture we have proposed. Following the exposed in Section 3.2, in the sense that our architecture might be implemented as an extension on systems requiring an audit characteristic, the prototype

follows a software scheme as a service using a microservice architecture. Figure 5 outlines the microservices and technologies used to implement our proposal. The implementation of each microservice includes Docker version 19.03.8 with Ubuntu 18.04 bionic operating system at the upper layer. Docker is a platform that delivers software in packages called containers. These containers are executed independently by the lightly operating system kernel and, therefore, use fewer resources than virtual machines, reference [52,53].

The following subsections describe how we have implemented the verifiability architecture, including its different parts; and the smart contracts within the blockchain.

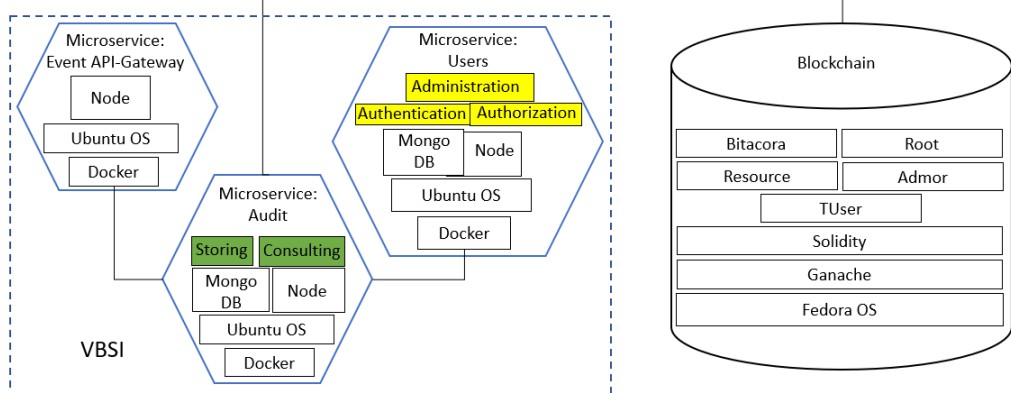

**Figure 5.** Microservices and technologies implemented in the proposal architecture.

### 6.1. VBSI

As you can see within the dotted rectangle in Figure 5, the VBSI part includes the following microservices: (a) Event API-Gateway; (b) Audit; and (c) Users.

Event API-Gateway was configured to accept HTTP methods (such as GET, POST, PUT, AND DELETE) in both ports 80 and 443. Port 80 was configured while developing the prototype, and port 443 to provide a private communication, implementing the transport layer security (TLS) protocol. This microservice is outward-facing for receiving external requests (such as the Internet); therefore, the rest of the ports were closed. Axios library was configured in the Event API-Gateway to deal with HTTP requests to the other microservices.

The *Users* microservice contains *Administration*, *Authentication* and *Authorization* services explained in Section 4.4. *User administration* involved to implement simple trusted tokens, which we have implemented with Nonce infrastructure (Nonce being an unguessable number built concatenating a random number with the current date); hashed messages were achieved with MD5-256 ( it is a hashing algorithm that takes an input string of any size and generates an output, which is a 256 bit size); and public key infrastructure with openPGP. *User-Authentication* was implemented using an access control password-based protocol;internally, a trusted compound token is generated when authentication is achieved. For *user authorization* we have implemented a compound token using Jason web token (JWT): https://jwt.io, accessed on 4 February 2022. All services were required to use HTTPS connections.

Green rectangles within the Audit microservice in Figure 5 illustrates the *Storing* and *Consulting* services explained in Sections 4.2 and 4.3. An example of storing and consulting service using Postman application will be explained in Section 7.3.

### 6.2. Blockchain and the Smart Contracts

The software component installed to execute the blockchain (as shown in Figure 5) was Ganache CLI v6.4.3. Ganache is a personal blockchain for rapid development and experimental tool under the Ethereum platform. It can be installed following the instructions shown in the Truffle suite at https://trufflesuite.com/ganache/index.html, accessed on 4 February 2022.

The smart contracts designed and shown in Figure 4 were implemented in Solidity language programming (Solidity available via https://docs.soliditylang.org/en/v0.7.4/, accessed on 4 February 2022, using version 6.12. The smart contracts can be downloaded at https://github.com/UPclopezpProjects/VBSI_smartContracts, accessed on 4 February 2022.

Library web3.js was used to connect with the blockchain. Unlike DAuth, mentioned in [54] where MetaMask extension was used in the client side description, our client side application is the *User* microservice, which, running node.js, connects with the blockchain by using an OAuth API web3 authentication protocol.

## 7. Case Study: An Avocado Cloud System

The following case study refers to an avocado supply chain system; although it could be adapted to any other, we provide this example to give a general idea of how it is integrated with the VBSI architecture we have proposed. Figure 6 illustrates such an integration, as one can see, it is a specialization of Figure 1.

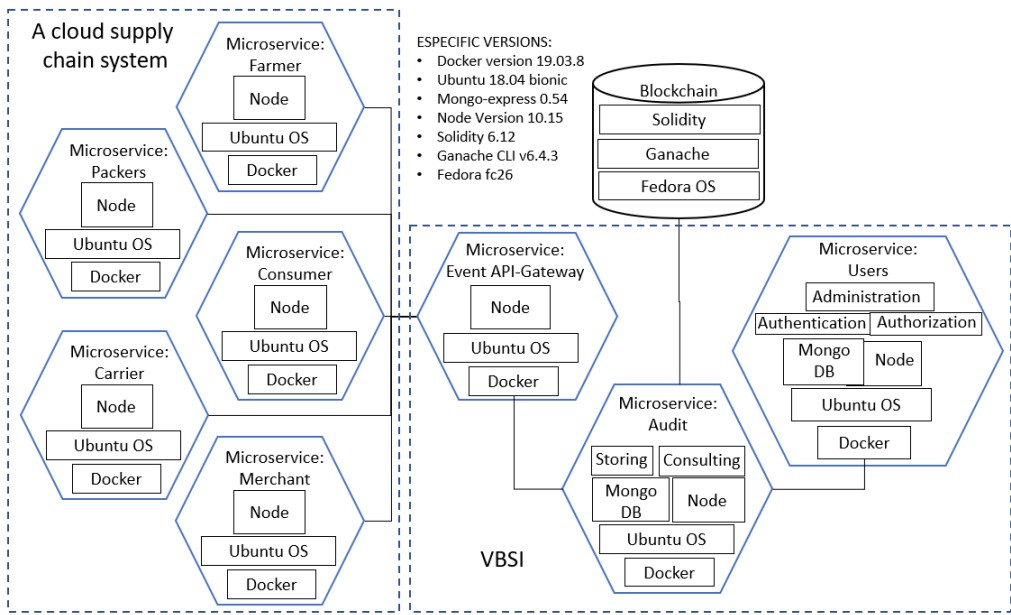

**Figure 6.** An avocado cloud system integrated with VBSIarchitecture.

### 7.1. Stages of the Supply Chain

The main requirement of the avocado supply chain is to register all events triggered from the producer to the end consumer. The stages of the supply chain are represented as individual microservices in the left part of Figure 6. Note that our experiments show that the configured operating system is similar to those used in VBSI, although it could be any other. The stages explanation, shown in the left part of the figure are:

1. *Farmer* can have one or more orchards, might sell his/her harvest, and be sent to packers.
2. *Packers* receive the avocado lot directly from the producer. They verify the lot, check regulatory documentation and establish if the fruit will be sent to be commercialized directly or a manufactured process.
3. *Carrier* are engaged in transporting the avocados from one point to the next in the chains of the supply chain.
4. *Merchant* buys the product to sell it directly to a consumer, a retailer (another merchant) or a wholesaler (another merchant).
5. *Consumer* is the user who buys the fruit. It is the last stage of the chain.

### 7.2. The Data Integrity Storing Process in the Experimental System

The transaction users interact at different stages with the avocados and every interaction records information to provide traceability. Then, the system can monitor the avocados from each stage. Figure 7 illustrates a sequence diagram that, at the top, shows the operations, at the bottom the created user's type, at the center the smart contracts created after the operations and at the right part (last column) the resources generated.

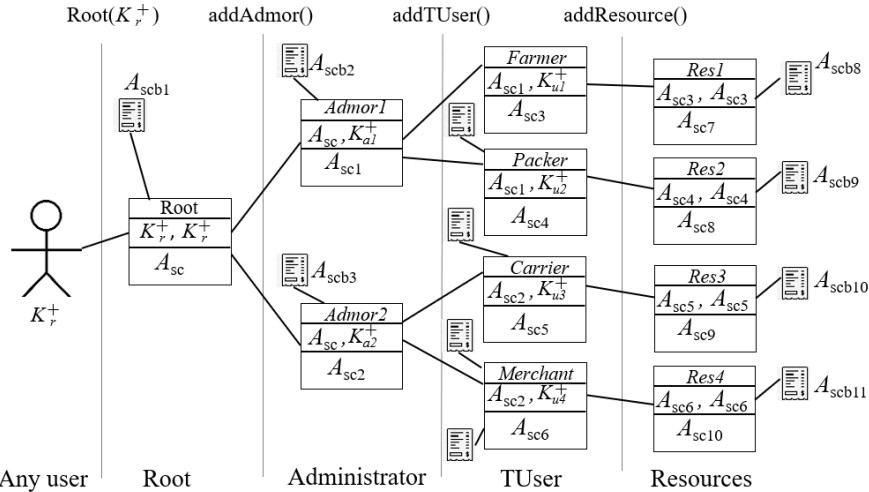

**Figure 7.** Sequence diagram: stating an execution tree operations of the avocado supply chain with their smart contracts.

The smart contracts in the figure are represented with a rectangle split in three parts: at the top with its name; in the middle with two parts: the smart contract used to create the new smart contract and the user key; at the bottom the address of such smart contract. The figure also states a tree smart contract instance from the root creation until the generation of users resources, the general steps are as follows:

- *Root creation:* following the protocol in [36] the Root user is created with key $K_r^+$. $A_{sc}$ is the smart contract address generated in the blockchain; and $A_{scb1}$ is the Root's Bitacora.
- *Administrators creation:* root, through smart contract $A_{sc}$, executes method $addAdmor(A_{sc}, K_{an}^+)$ to create two administrators (*Admor1* and *Admor2*); $K_{an}^+$ in this case are $K_{a1}^+$ and $K_{a2}^+$; and the smart contracts generated are $A_{sc1}$ and $A_{sc2}$ respectively. Their smart contracts Bitacoras are $A_{scb2}$ and $A_{scb3}$
- *TUsers creation:* the explanation of creating a TUser by an administrator is similar to that explained by the root creating administrators. In this case *Admor1* creates a *Farmer* and a *Packer* user; and *Admor2* creates a *Carrier* and a *Merchant* user.
- *Resources creation:* resources are added via a TUser by executing method $addResource(string)$. The figure shows resources created with addresses from $A_{sc7}$ to $A_{sc10}$. These resources will be used to manage harvests. For example, *Farmer* will register new hashed data of the harvest in the blockchain to start the traceability; *Carriers* will register the harvest transportation from one part to another; *Packer* will create a new harvest batch with the information reported by the *Farmer* and will carry out the selection procedures; *Merchant* will register data information about when the lot was received and when it is sent to the wholesalers, retailers or ready to be sell for the consumers. In each stage, a QR code is generated, containing the transaction address generated in each stage. The QR code might be used by the *consumer* or any user of each stage of the supply chain to obtain the traceability and verify the origin of the avocado.

Figure 8 shows a dummy representation of the smart contracts executed in Figure 7. Note that each block contains three transactions represented by each of the smart contracts. This figure will be related with the below explanation of Figure 9 in the next subsection.

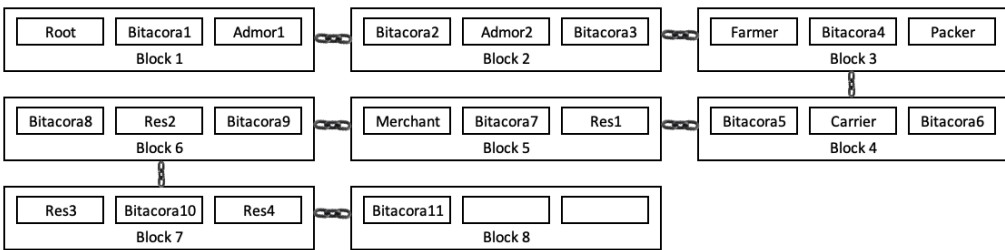

**Figure 8.** Blockchain representation of the smart contracts shown in Figure 7.

### 7.3. The Integrity Verifiability Process

Figure 9 illustrates a summary of those resources created and shown in Figure 7, applying the *Storing* service explained in Section 4.2 and following the experimental storing process of Section 7.2. The Figure 9 illustrates two tables. The left one shows some data of the resources stored via off-chain. In particular, this table shows seven columns: *Stage* is the name of the stage in the supply chain; *Id* is used to identify the row; *R* is the receipt received from the blockchain after a *Storing* process is called; *D* is the data to be hashed; $Hash(D)$ is the hashed data; *Asc* and *Atran* are the transaction and smart contract addresses where the row were stored within the blockchain.

| Resources | | | | | | | | Bitacora | | | | | | | | | |
|---|---|---|---|---|---|---|---|---|---|---|---|---|---|---|---|---|---|
| Stage | Id | R | D | Hash(D) | Asc | Atran | | Ablock | Asc | Op | Asig | When | Operation type | Where | Target | To | Hx |
| Farmer | 1 | r1 | d1 | h1 | Ascb8 | t1 | | 3 | Ascb8 | POST | S1 | w1 | FarmerCreation | IP1 | Server | To1 | h1 |
| Packer | 2 | r2 | d2 | h2 | Ascb9 | t2 | | 3 | Ascb9 | POST | S2 | w2 | PackerCreation | IP2 | Server | To2 | h2 |
| Carrier | 3 | r3 | d3 | h3 | Ascb10 | t3 | | 4 | Ascb10 | POST | S3 | w3 | CarrierCreation | IP3 | Server | To3 | h3 |
| Merchant | 4 | r4 | d4 | h4 | Ascb11 | t4 | | 5 | Ascb11 | POST | S4 | w4 | MerchantCreation | IP4 | Server | To4 | h4 |

**Figure 9.** The **Left** table shows some data about where the resources are stored off-chain and in the blockchain; and the **right** table shows the logs stored in each Bitacora of the smart contracts.

On the other hand, the right table of Figure 9 shows some information stored within the smart contracts, which guarantees integrity by a verification procedure as explained in Section 4.3. This table shows ten columns (see Table 4 to reference for the meaning of their titles): the first one, *Ablock*, specifies the block number where the smart contract is located within the blockchain, check Figure 8 to link it; in this case all event methods (*Op* in the table) triggered by the requesters were POST (because events were the creation of the stages, see Table 2 to collate), column *When* is denoted with unix epoch number, *Operation type* being the operation description carried out; column *Where* is the IP number from where the request was triggered; *To* is the session user identifier; and *Hx* the hashed data.

Table 5 shows a representative values of variables included in Figure 9, assuming that the rest variables can be interpreted in a similar way.

Once explained Figures 7–9 about how the logs of events are stored in the blockchain through the smart contracts, and exemplifying some values in Table 5; we are ready to instance a consulting operation as demonstrated in Section 4.3. Assume that the Event API-Gateway receives a consulting operation request (it might be from any microservice of the supply chain as illustrated in Figure 6); an example of the consulting request is shown in the right part of Figure 10, note that the data to be sent are $r1$ and $h1$ of Table 5, these values are stored via off-chain (see the left table of Figure 9). Taking into account that $h1$ coincides with the stored information in the blockchain (see the right table of Figure 9) the answer will be $Res = \{\!|true, r1|\!\}$, pinpointing that $h1$ is valid for $r1$. If $h1$ would have been modified, then the answer would be $Res = \{\!|false, r1|\!\}$.

**Table 5.** Representative variable values of Figure 9.

| Abbreviation | Description |
|---|---|
| r1 | {*To* = "eda1917110fb1ea22709138e38ab9f0", $A_{tr}$ = "0x586382da3ef2a8026738123ca47d656943bdb573af6c185b192492ec3c29d4b6", $A_{sc}$ = "0x1F4DD9f716bbb9D4b2FdA10D2C7a7D6E2C90580d"} |
| d1 | {name = "FarmerName", stage: "Farmer", gas = "900000", pass = "sa23lfd_2", key = "0xCd801D62AF617641964db500D98146eFCEF610E0"} |
| h1 | 6d64619ba21d1114facb3efd54a1d4be |
| Ascb8 | 0x1F4DD9f716bbb9D4b2FdA10D2C7a7D6E2C90580d |
| t1 | 0x586382da3ef2a8026738123ca47d656943bdb573af6c185b192492ec3c29d4b6 |
| s1 | c5a13ecf53fb22134a4613120da887e4 |
| w1 | 1638991308 |
| IP1 | 189.129.78.230 |
| To1 | eda1917110fb1ea22709138e38ab9f0 |

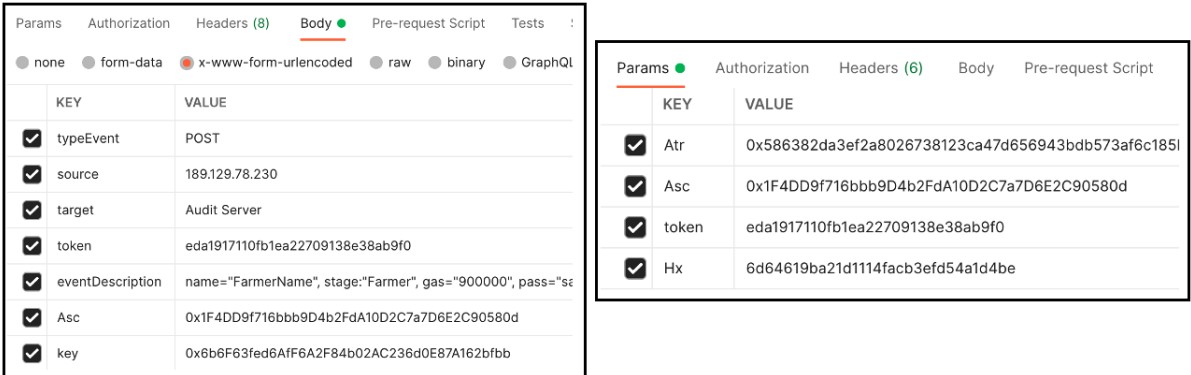

Storing　　　　　　　　　　　　　　　Consulting

**Figure 10.** Storing and consulting service example using Postman application.

## 8. Validation and Proofs

This section provides some validations to our prototype and the case study to give certainty about its viability. They consist in setting two types of agents: friendly and unfriendly. Both types of agents, acting their corresponding roles, attempt to execute transactions. Each of the executed transactions must be registered within the blockchain and must generate a log.

We explain the behavior of the agents, some results about their implementations; a data integrity verifiability example interface; and the system complexity by stating a run showing the latency and processing time of the system.

### 8.1. Friendly and Unfriendly Agents

Friendly agents consist of a set of compliant agents following the rules of expected behavior. This type of agent aims to identify programming errors under normal situations. These situations mean that if a request contains *n* fields, and each field is configured with a specific type, then friendly agents send *n* fields without changing the types. On the other hand, unfriendly agents are non-compliant. They do not follow the rules, can change types, and change an unexpected behavior. Table 6 describes the behavior of the unfriendly agents.

**Table 6.** Behaviour of the unfriendly agents.

| Resource | Unfriendly Agent Description |
|---|---|
| Create Root | We replicated the proofs executed in [36], but adapted in this context. A set of agents do not follow the protocol rules for the root creation. |
| Create Administrators and TUsers | They might change the permissions, the tokens value, and change the types of the fields. |
| Add data to stages | They can try to add data to stages not permitted. |

Currently, there exist tools that can help with the software development process as a tester in the requests, e.g., the Postman application, which we also used. However, to implement the behavior of Table 6, was necessary to develop a tool with Java threads using an interface as shown in Figure 11. At the top of the figure can be seen some *Resources* of the first column of Table 6, among other things. Each tab of the figure was configured to execute automatic or manual requests. The number of requests must be input and the amount of friendly and unfriendly agents (honest or dishonest in the figure). Depending on the number *N* of agents, internally, the simulator creates *N* threads acting the corresponding role of the agents. Each friendly request must send the data as you can see in the input fields of the figure; note that some fields are required to be introduced manually, and others follow a regular expression to calculate random and automatic data fields.

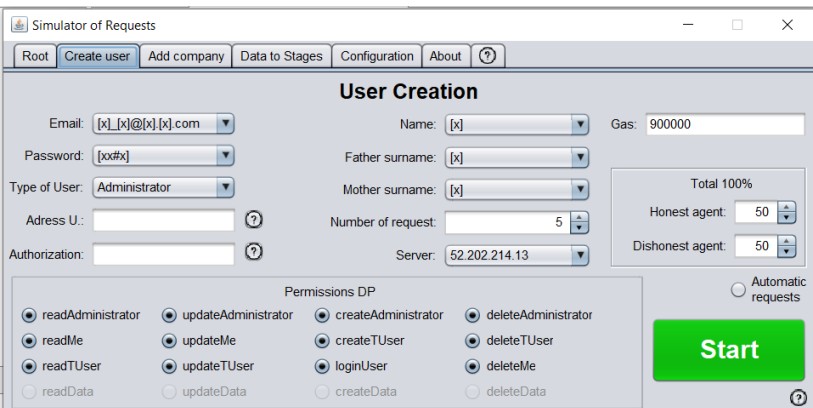

**Figure 11.** User interface to test the creation of different types of users.

At the end of each run, the system generates a detailed description as you can see in Figure 12. A version of the tool can be downloaded from GitHub: https://git.io/JDtin, accessed on 4 February 2022.

We executed multiple stress proofs with this customized application and configured a malicious scenario. The following list describes the aspects of this tool and the prototype, which helped in the design of our proposal:

- Programming: we corrected some programming errors. It involved fixing from simple validations during the development process until more complex troubles, such as distributed and concurrent programming that sometimes are very difficult to detect.
- Adjustment to the smart contracts: although our smart contracts are illustrated as class diagrams to give a general representation, they were implemented on solidity programming language.
- Adjustment to the architecture: developing the prototype and the tester tool was more clear to make some adjustments to the proposed architecture.

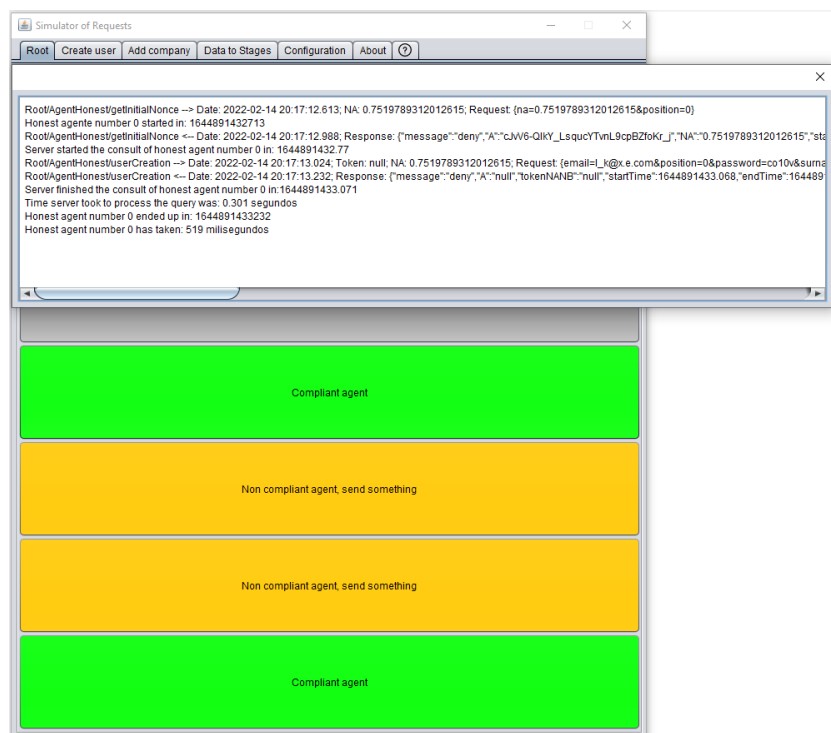

**Figure 12.** Latency details shown in the graphical interface while sending and receiving requests.

### 8.2. A Data Integrity Verifiability Example Interface

Figure 13 shows the outstanding interfaces from a mobile application we developed to verify the integrity of data previously stored within a blockchain. The figure contains four images, from left to right: (1) Login interface: where the user might sign in by a previous register or by social networks; (2) A menu option with the possibility to scan QR codes to see the traceability of the avocado supply chain that the user wants to buy; (3) The traceability interface with a specific location in Google maps, and options such as stage details forward and backward arrow navigation, among others; and (4) The evidence log taken from the blockchain illustrates the interface that the final user can see to verify the details stored in the blockchain, otherwise, such details would not be shown.

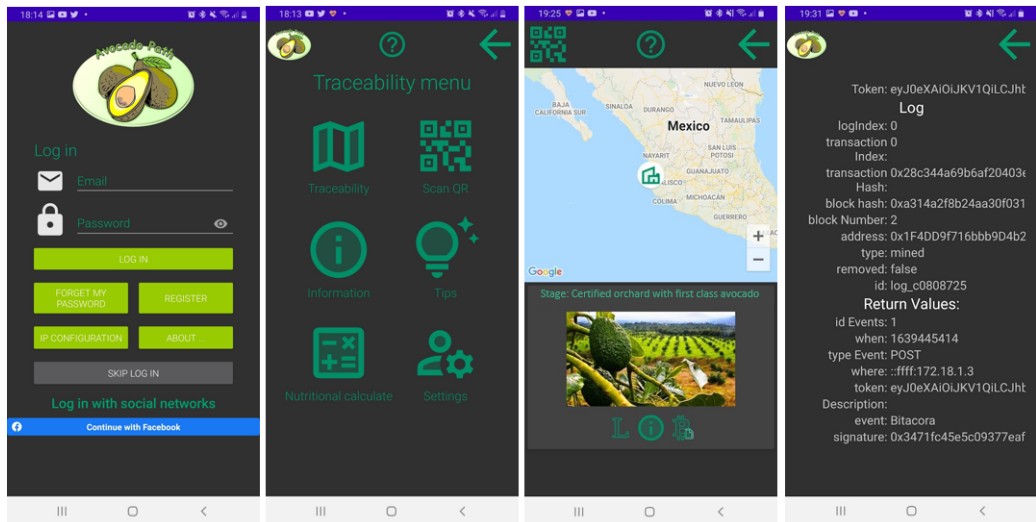

**Figure 13.** Mobile application to check the traceability of an avocado supply chain and the log example obtained from the blockchain.

*8.3. System Complexity*

We carried out some tests with respect to the amount of resources required to run the storing and consulting services. The tests were latency and processing time. Latency is the time any supply chain microservice takes to reach its destination and return, taking into account an HTTP request. The processing time is the time in milliseconds (ms) that the API Gateway takes to execute a request since it arrives until it is processed and returned to the requester. Table 7 shows some details of one of the runs. The first column states the stage of the cloud supply chain microservice from where the request was established. The following three columns state the transaction latency (*Start*, *End*) that the request takes (*Time*) to reach its destination and return to the microservice. The fifth column pinpoints the HTTP method executed (POST, PUT, DELETE, or GET). The next three columns state the processing time within the Event API-Server. The last column is a brief description about the type of request. Note that *Start* and *End* columns are expressed in epoch Linux time, and column *Time* is expressed in milliseconds, being it the difference between *End* and *Start*.

**Table 7.** Transaction latency from a Cloud Supply Chain and processing time in the Event API-Gateway.

| Cloud SChain | Latency from a Cloud Supply Chain | | | HTTP Method | Processing Time | | | Description |
|---|---|---|---|---|---|---|---|---|
| | Start | End | Time (ms) | | Start | End | Time (ms) | |
| Farmer | 1645057568522 | 1645057575176 | 6654 | POST | 1645057575177 | 1645057577016 | 1839 | Creating a Farmer |
| Farmer | 1645057572394 | 1645057575191 | 2797 | GET | 1645057575192 | 1645057576297 | 1105 | Consulting a log |
| Carrier | 1645057569135 | 1645057575901 | 6766 | PUT | 1645057575902 | 1645057577691 | 1789 | Updating information |
| Carrier | 1645057574510 | 1645057576801 | 2291 | GET | 1645057576802 | 1645057577797 | 995 | Consulting a log |
| Packers | 1645057570272 | 1645057576902 | 6630 | DELETE | 1645057576903 | 1645057578528 | 1625 | Deleting information |
| Packers | 1645057574996 | 1645057577904 | 2908 | GET | 1645057577905 | 1645057578795 | 890 | Consulting a log |
| Merchant | 1645057570477 | 1645057577011 | 6534 | POST | 1645057577012 | 1645057578832 | 1820 | Creating a Merchant |
| Merchant | 1645057575175 | 1645057577365 | 2190 | GET | 1645057577366 | 1645057578474 | 1108 | Consulting a log |
| Consumer | 1645057576367 | 1645057578762 | 2395 | GET | 1645057578763 | 1645057579562 | 799 | Consulting a log |

We can analyze that methods POST, PUT, and DELETE, which correspond to storing information within the blockchain, are very similar and took 6646 ms latency on average, respectively, a 1768 ms processing time. On the other hand, consulting information took 2516 ms latency on average, respectively 979 ms in processing time. These statistics could oscillate using another blockchain platform. The average transmission rate from the microservices was a 1.91 Mbps upload and 29.78 Mbps download.

## 9. Discussion and Conclusions

The work presented in this article is closely related with [35,36]; in this section, firstly, we discuss the main differences with our work, highlighting the extensions and our main contributions, then we give the conclusions.

*9.1. Discussion*

Table 8 shows, with a ✓, the main characteristics supported in the works [35,36] and our proposal model. We coincide in both works in prevailing characteristics of data integrity, blockchain technology, auditing and traceability services, and implementing the microservice architecture. However, reference [35] focuses exclusively to supply chain systems resulting in very bounded implementations, so, in this work, we decide to include a more generalized solution for cloud systems. In contrast with [35], our proposal starts with the security suggestion of [36] concerning the auditing mechanism from when a Root user is created in a decentralized environment.

On the other hand, we propose a decentralized architecture that extends both works [35,36], presenting a general solution for cloud computing services; and considering



the creation of several types of users and roles; respectively, making scalable and modular the audit mechanism proposed. Thus, we can integrate more services through smart contracts (as shown in Figure 4) and traceability with the processing time and latency shown in Section 8.3. A critical remark is that we can provide information by consulting the log stored in the blockchain provided by the smart contracts about *how* an event was generated, *who* generated it, *what* included the event, *where* was it generated, and *when* was the event generated.

**Table 8.** Main characteristics of the following works [35,36] vs. the proposal model.

| No. | Characteristics | [35] | [36] | The Proposal Model |
|-----|-----------------|------|------|--------------------|
| 1 | Provides data integrity | ✓ | ✓ | ✓ |
| 2 | Uses blockchain technology | ✓ | ✓ | ✓ |
| 3 | Focuses exclusively to supply chain systems | ✓ | | |
| 4 | Focuses to generalized cloud systems | | ✓ | ✓ |
| 5 | Provides auditing service | ✓ | ✓ | ✓ |
| 6 | Provides traceability service | ✓ | ✓ | ✓ |
| 7 | Involves different role users | ✓ | | ✓ |
| 8 | Integrity: Stores events as hashed messages | ✓ | | |
| 9 | Integrity: stores events as logs | | ✓ | ✓ |
| 10 | Integrity: stores hashed messages in logs | | ✓ | ✓ |
| 11 | Implements microservice architecture | ✓ | ✓ | ✓ |
| 12 | Ensures the creation of a root user | | ✓ | ✓ |
| 13 | Ensures the creation of different types of users | | | ✓ |

### 9.2. Conclusions

Organizations tend to move their information systems to the cloud. Increasingly, one of the questions from users is about the integrity of storing data. Thus, providing data integrity is an essential activity that adds value to organizations. If this activity is provided using the appropriate technologies, it generates more value.

In this paper, we described a data integrity verifiability architecture comprised of the following:

(a) An event API-gateway able to receive HTTP requests and to decide if it is addressed to an administrative user, for storing hashed data, or for consulting (audit) issues;

(b) An administrative user service, able to register different types of users, authentication service able to provide tokens and authorization service, able to decide the permissiveness of the requester;

(c) A storing service that saves hashed messages that later will be used as an audit process; and

(d) A consulting mechanism that provides certainty about the integrity of previously stored data.

We described each of the components, and to give a better idea about implementations details, we supplied a prototype description. We also developed a pair of tools to validate our proposal, consisting of a tester tool and a mobile application to be used for end-users. In addition, as shown in this research, a distributed system connected to a public blockchain induces high transaction times due to processing and, hence, a longer latency.

This architecture can be replicated in different applications that do not natively have blockchain technology built-in as an extension for those systems requiring modular and scalable audit characteristics. Our solution has been built using a microservice architecture because, currently, this approach is the tendency in the software development area.

This is an ongoing work, we are looking for performance statistics, and it has been implemented under the Ethereum platform. Still, it could be interesting to compare other platforms, for example, Hyperledger.

**Author Contributions:** Conceptualization, J.C.L.-P.; data curation, J.C.L.-P. and L.A.M.-R.; formal analysis, J.C.L.-P. and L.A.M.-R.; funding acquisition, J.C.L.-P. and L.A.M.-R.; investigation, J.C.L.-P.; methodology, J.C.L.-P., L.A.M.-R. and I.A.-B.; project administration, J.C.L.-P.; Resources, J.C.L.-P., L.A.M.-R. and I.A.-B.; software, J.C.L.-P. and I.A.-B.; supervision, J.C.L.-P. and L.A.M.-R.; validation, J.C.L.-P., L.A.M.-R. and I.A.-B.; visualization, J.C.L.-P., L.A.M.-R. and I.A.-B.; writing—original draft, J.C.L.-P.; writing—review and editing, J.C.L.-P., L.A.M.-R. and I.A.-B. All authors have read and agreed to the published version of the manuscript.

**Funding:** This research was partly funded by COECyTJAL, issued in the FODECIJAL grant, code 8217-2019, under the fund to address state problems 2019; and it was partially funded by the Mexican National Council for Science and Technology (CONACYT) through research projects 882 and 613.

**Institutional Review Board Statement:** Not applicable.

**Informed Consent Statement:** Not applicable.

**Data Availability Statement:** The smart contracts can be downloaded at https://github.com/UPclo pezpProjects/VBSI_smartContracts, accessed on 4 February 2022.

**Conflicts of Interest:** The funders had no role in the design of the study; in the collection, analyses, or interpretation of data; in the writing of the manuscript, or in the decision to publish the results.

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
