# Peer review of "A Cloud Microservices Architecture for Data Integrity Verifiability Based on Blockchain"

_applsci, doi:10.3390/app12052754_

Round 1
Reviewer 1 Report
The paper is very interesting and deals with a very interesting topic.
However, there are some aspects that in my opinion should be improved before the publication.
1)In my opinion the improvement with respect to the literature is not sufficiently underlined.
2) Experimental results are not clear, they should be summarized in order to help the reader.
Minor
pictures with Spanish sentences should be replaced with English ones
Author Response
Reviewer 1 Comments
Open Review
(x) I would not like to sign my review report
( ) I would like to sign my review report
English language and style
( ) Extensive editing of English language and style required
( ) Moderate English changes required
( ) English language and style are fine/minor spell check required
(x) I don't feel qualified to judge about the English language and style
Yes |
Can be improved |
Must be improved |
Not applicable |
|
Does the introduction provide sufficient background and include all relevant references? |
( ) |
( ) |
(x) |
( ) |
Is the research design appropriate? |
( ) |
(x) |
( ) |
( ) |
Are the methods adequately described? |
( ) |
( ) |
(x) |
( ) |
Are the results clearly presented? |
( ) |
(x) |
( ) |
( ) |
Are the conclusions supported by the results? |
(x) |
( ) |
( ) |
( ) |
Comments and Suggestions for Authors
The paper is very interesting and deals with a very interesting topic.
However, there are some aspects that in my opinion should be improved before the publication.
Response to Reviewer 1 Comments
Font code to understand the main changes in the article:
Changes: highlighted red text
1)In my opinion the improvement with respect to the literature is not sufficiently underlined.
Response: Thanks for this suggestion.
We have improved all section 2, Preliminaries and Related Work, highlighted with red font. In general, we have added 4 new cites [14, 18, 35, 55].
2) Experimental results are not clear, they should be summarized in order to help the reader.
Response: Thanks for this suggestion.
We have modified section 7, in particular, re-structured sections 7.2 and 7.3 by including more details and explanations about the storing and consulting process. In addition, we have added a new subsection 8.3, which we have titled “System complexity”, in which we explain the latency and processing time that our prototype takes; the subsection includes a new Table 8, showing one of the runs executed in our prototype. With these new changes, we have updated our conclusions (highlighted in red font) to summarize more clearly our experimental results.
Minor
pictures with Spanish sentences should be replaced with English ones
Response: Thanks for this observation. We have replaced Figures 11, 12, and 13; right now, it contains English texts completely.
Additionally, the following is a summary of the changes carried out throughout the manuscript, we have:
- Highlighted the architecture features in the last sentence of the abstract.
- Added cite [14] in the Introduction part (second paragraph, page 2) to describe the different use cases where blockchain and smart contracts are applied.
- Improved the introduction section, highlighting the open problem concerning microservices, connecting it with the aim of the paper.
- Improved all section 2, Preliminaries and Related Work, which has been highlighted with red font. In general, we have added 4 new cites [14, 18, 35, 55].
- Included the last paragraph in section 2.3 to explain in more detail Table 1 and the main extensions of our work (highlighted with red font)
- Included a column description (so-called Formula) in Table 2, pinpointing the number of the equation where the abbreviation can be consulted. In addition, we add the equation numbers of each of the formulas, as you can see in the formulas (1) to (10) in section 4.
- Modified in section 4.4.3 the JSON language code's presentation and adding a description to be more clear and understandable.
- Added a description about the meaning of MD5-256. It was put as a footnote 2 in section 6.1
- Included more details in Section 6.2: including instructions details about the ganache implementation; and the authentication procedure, including the suggested research DAuth.
- Re-structured sections 7.2 and 7.3 by including more details and explanations about the implementation of the main services provided for our architecture.
- Added a new subsection 8.3, which we have titled "System complexity". We explain the latency and processing time that our prototype takes; the subsection includes a new Table 8, showing one of the runs executed in our prototype. With these new points, we have updated our conclusions (highlighted in red font).
- Precise the penultimate paragraph of the conclusion to specify the application area of our work.
- Re-structured sections 7.2 and 7.3 by including more details and explanations about the storing and consulting process. In addition, we have added the last paragraph in section 7.3, which explains how the data integrity process is validated. Figure 10 previously was in section 6.1; now, it is included in section 7.3 with its graphical explanation to strengthen this section.
- Replaced Figures 11, 12, and 13; right now, they contain English texts completely.

Reviewer 2 Report
Originality/Novelty: The authors work on microservices data integrity verification using blockchain. The paper contains are good and well structure with contribution to the existing knowledge.
Significance: The use od smart contract make the system autonomous and security verification can be view with the use of blockchain immutable log file.
Quality of Presentation: The overall presentation and structure of the paper is well handle by the authors.
Scientific Soundness: The authors propose protocols is implemenataion needs to be more clarity. Regarding the platform use , system complexity and computation complexity when smart contract are deployed in blockchain paltform
Interest to the Readers: The authors must includes some more recent work along with some applications area where this protocols may be deploy.
The following comments are need to address to improve the paper:
- Abstract : The last line need to be more specific in term of contribution like which feature make your proposed system more suitable to the cloud.
- Introduction: Mention your objectives and gap to build up the problem ststement to your contribution.
- section 2.2. Data integrity in cloud systems, first two line look vague .
- section 2.3, authors mention paper reference number 33 and 34 as close to their contribution paper, these two papers are their previous published work. so they must explain what is extention they made in this paper ??
- section 4, non of the equations are number or reference in the text. also in 4.4.3. Authorization one piece of code is written without any title . this is pseudo code or smart contract or function ???
- section 6.1. VBSI, It is mention that hashed messages was achieved with md5-256, what is md5-256 ??
- In section 6, authors talk about ethereum platform use as a blockchain environment, but details are missing. Please see even include these paper regarding Ganache and smart contract for etherum platform . a) DAuth: a decentralized web authentication system using Ethereum based blockchain. b) An Overview of Smart Contract and Use Cases in Blockchain Technology
- The verification section data integrity part explained need more clarity how proposed protocol maintained data integrity ?
Author Response
Reviewer 2 Comments
Open Review
(x) I would not like to sign my review report
( ) I would like to sign my review report
English language and style
( ) Extensive editing of English language and style required
( ) Moderate English changes required
(x) English language and style are fine/minor spell check required
( ) I don't feel qualified to judge about the English language and style
Yes |
Can be improved |
Must be improved |
Not applicable |
|
Does the introduction provide sufficient background and include all relevant references? |
( ) |
(x) |
( ) |
( ) |
Is the research design appropriate? |
(x) |
( ) |
( ) |
( ) |
Are the methods adequately described? |
(x) |
( ) |
( ) |
( ) |
Are the results clearly presented? |
( ) |
(x) |
( ) |
( ) |
Are the conclusions supported by the results? |
(x) |
( ) |
( ) |
( ) |
Comments and Suggestions for Authors
Originality/Novelty: The authors work on microservices data integrity verification using blockchain. The paper contains are good and well structure with contribution to the existing knowledge.
Significance: The use od smart contract make the system autonomous and security verification can be view with the use of blockchain immutable log file.
Quality of Presentation: The overall presentation and structure of the paper is well handle by the authors.
Response to Reviewer 2 Comments
Font code to understand the main changes in the article:
Changes: highlighted red text
Scientific Soundness:
The authors propose protocols is implemenataion needs to be more clarity.
Response: We have re-structured sections 7.2 and 7.3 by including more details and explanations about the implementation of the main services provided for our architecture.
Regarding the platform use , system complexity and computation complexity when smart contract are deployed in blockchain paltform
Response:
We have added a new subsection 8.3, which we have titled “System complexity”, in which we explain the latency and processing time that our prototype takes; the subsection includes a new Table 8, showing one of the runs executed in our prototype. We have updated our conclusions with these new points (highlighted in red font).
Interest to the Readers: The authors must includes some more recent work along with some applications area where this protocols may be deploy.
Response: Thanks for this suggestion.
We have improved all section 2, Preliminaries and Related Work, highlighted with red font. In general, we have added 4 new cites [14, 18, 35, 55].
In addition, we have precise the penultimate paragraph of the conclusion to specify the application area of our work.
The following comments are need to address to improve the paper:
- Abstract : The last line need to be more specific in term of contribution like which feature make your proposed system more suitable to the cloud.
Response: We have highlighted the architecture features in the last sentence of the abstract.
- Introduction: Mention your objectives and gap to build up the problem ststement to your contribution.
Response: We have improved the introduction section, highlighting the open problem concerning microservices, connecting it with the aim of the paper.
- section 2.2. Data integrity in cloud systems, first two line look vague .
Response: Effectively we put a typo with a word, and it made the paragraph unclear. We have paraphrased it to be more understandable (highlighted with red font).
- section 2.3, authors mention paper reference number 33 and 34 as close to their contribution paper, these two papers are their previous published work. so they must explain what is extention they made in this paper ??
Response: Thanks for this comment. We have included the last paragraph in section 2.3 (highlighted with red font), which explains in more detail Table 1 and the main extensions of our work. It is important to clarify that Table 1 previously was at the beginning of section 3, but we think that this change is better to do more clear our contribution and extensions.
- section 4, non of the equations are number or reference in the text.
Response: Thanks for this comment. In Table 2, we have included a column description (so-called Formula) pinpointing the number of equation where the abbreviation can be consulted. In addition, we have added the number of equations of each of the formulas as you can see from formula (1) to (10) in section 4.
also in 4.4.3. Authorization one piece of code is written without any title . this is pseudo code or smart contract or function ???
Response: In this section 4.4.3, we have modified the presentation of the JSON language code and adding also a description to be more clear and understandable.
- section 6.1. VBSI, It is mention that hashed messages was achieved with md5-256, what is md5-256 ??
Response: We have added a description about the meaning of MD5-256. It was put as a footnote 2 in section 6.1
- In section 6, authors talk about ethereum platform use as a blockchain environment, but details are missing. Please see even include these paper regarding Ganache and smart contract for etherum platform . a) DAuth: a decentralized web authentication system using Ethereum based blockchain. b) An Overview of Smart Contract and Use Cases in Blockchain Technology
Response: Thanks for this suggestion.
We have included more details in Section 6.2: including instructions details about the ganache implementation; and the authentication procedure, including the suggested research DAuth.
In addition, we have added cite [14] in the Introduction part (second paragraph, page 2), which describes the different use cases where blockchain and smart contracts are applied.
- The verification section data integrity part explained need more clarity how proposed protocol maintained data integrity ?
Response: Thanks for this suggestion.
We have re-structured sections 7.2 and 7.3 by including more details and explanations about the storing and consulting process. In addition, we have added the last paragraph in section 7.3, which explains how the data integrity process is validated. Figure 10 previously was in section 6.1, it was moved to section 7.3 to strengthen the section with its graphical explanation.

Round 2
Reviewer 1 Report
Authors follow the review instructions to improve the paper and for this reason I suggest the acceptance
Author Response
Dear reviewer,
Thanks for your suggestions we are convinced that new updates in the paper make it a fined version. The academic editor suggested some changes, the following list abstracts about it:
- Re-proofread the manuscript and corrected some grammatical errors throughout the document.
- Eliminated different extra spaces throughout the document, and corrected the typos suggested and the new ones we found.
- Added a new paragraph (lines 45 to 52) to be more clear with respect to the innovation of the paper and give a better understanding of its scope. In addition, we have added a new subsection 9.1 named Discussion where we enforce more this aspect and we explain the main difference concerning works [36] and [37]. The sub-section abstracts our main innovations and includes Table 8, which previously was at the end of sub-section 2.3. We think this change is better because after explaining our complete architecture, the reader is ready to understand the differences concerning works [36] and [37].
- Delimited our work highlighting it in lines 51 and 52.
- Answered the details of the blockchain implementation in section 6.2.
- Added a new cite [21] to enforce the meaning of office systems. We modified the paragraph (lines 98-102) to be more clear about why we are introducing an evolutionary context of the strategies in the software development process starting with office systems.
- Restructured various sentences in lines: 29 to 32; 37 to 40; 66 to 70; 51 to 55; 93 to 97; 95 to 96; 98-102; 104-111; 131 to 133.
Thanks again.
Juan Carlos López Pimentel
Corresponding author
Universidad Panamericana. Facultad de Ingeniería. Álvaro del Portillo 49, Zapopan, Jalisco, 45010, México.
Email: clopezp@up.edu.mx Phone: +52 (961) 1883103